# *Leishmania* Infection during Ruxolitinib Treatment: The Cytokines-Based Immune Response in the Setting of Immunocompromised Patients

**DOI:** 10.3390/jcm12020578

**Published:** 2023-01-11

**Authors:** Andrea Duminuco, Salvatore Scarso, Alessandra Cupri, Nunziatina Laura Parrinello, Loredana Villari, Grazia Scuderi, Giuliana Giunta, Salvatore Leotta, Giulio Antonio Milone, Giulia Giuffrida, Giuseppe Alberto Palumbo, Giuseppe Milone

**Affiliations:** 1Division of Hematology with BMT, A.O.U. Policlinico “G.Rodolico-San Marco”, Via S. Sofia 78, 95123 Catania, Italy; 2National Center for Global Health, Istituto Superiore di Sanità, 00161 Rome, Italy; 3Unit of Pathological Anatomy, A.O.U. Policlinico “G.Rodolico-San Marco”, Viale Carlo Azeglio Ciampi, 95123 Catania, Italy; 4Division of Hematology with BMT, Istituto Oncologico del Mediterraneo, 95029 Viagrande, Italy; 5Department of Scienze Mediche Chirurgiche e Tecnologie Avanzate “G.F. Ingrassia”, University of Catania, 95123 Catania, Italy

**Keywords:** *Leishmania*, ruxolitinib, myelofibrosis, immune response, cytokines

## Abstract

Ruxolitinib is a JAK1/2 inhibitor that has revolutionized the approach to myelofibrosis. On the one side, this drug can rapidly improve the symptoms related to the hematological disease; on the other side, the inhibition of JAK1/2 can lead to immunosuppression which may increase the risk of infections, due to a change in the cytokine balance in favor of anti-inflammatory cytokines, to direct inhibition of immune cells, and to the suppression in the production of specific antibodies. In this patient setting, much is known about possible viral and bacterial infections, while little is reported in the literature concerning parasitic infections, specifically leishmaniasis. *Leishmania* is a parasitic infection that can cause serious problems in immunosuppressed patients. The parasite can invade the bloodstream and cause a wide range of symptoms, including fever, weight loss, and anemia. In severe cases, it can lead to multi-organ failure and, rapidly, death. Early diagnosis and prompt treatment are essential especially for these patients, unable to respond adequately. In this case and the following review of the existing literature, the cytokine kinetics and the production of specific anti-*Leishmania* antibodies represent characteristic aspects capable of providing a more in-depth understanding of the mechanisms underlying these complex clinical cases in an immunocompromised patient.

## 1. Introduction

Myelofibrosis (MF) is a chronic hematological disease characterized by heterogeneous manifestations and a highly variable prognosis, with an overall survival that may range from <2 to 20 years [1], depending on the prognostic factors at the time of diagnosis and their variations during the evolution of the disease. The crucial role of driver mutations such as JAK2, MPL, and CALR has changed our understanding and dealing with MF. In this regard, the availability of drug inhibitors of Janus kinase has revolutionized the MF therapeutic approach, ensuring a rapid improvement of systemic symptoms, and increasing of overall survival. For these reasons, ruxolitinib (a JAK2 inhibitor) is often chosen as the first-line treatment in MF [2]. It is widely known that ruxolitinib wields mighty anti-inflammatory and immunosuppressive effects. Dendritic, natural killer, T-helper, and regulatory T-cells are comprehended targets of ruxolitinib, able in this way to act on both the innate and adaptive immune system. [3]. The possible response of the human immune system in contrast to infections in patients undergoing ruxolitinib treatment is still an open challenge that needs further clarification.

## 2. Case Presentation

We report the case of a 60-year-old female who was referred to our Center four years before for splenomegaly (4 cm below lower costal margin), with moderate anemia (Hb 9.2 g/dL), and leukopenia (WBC 3000/mmc with 1320/mmc neutrophils). Platelets were in the range (263,000/mmc). Bone marrow evaluation showed hyperplasia of the granule- and megakaryopoiesis, blast cells <5%, and grade 2 fibrosis. Primary myelofibrosis diagnosis was made, IPSS score 1 (Intermediate-1 risk class for anemia, with a median survival of 7.9 years) [4]. Follow-up was started with sporadic erythropoietin treatment. After two years, she reported a worsening in hemoglobin value (7.8 g/dL, requiring transfusion support), leukocytosis (WBC 19,500/mmc), and systemic symptoms (night sweating). Bone marrow biopsy confirmed grade 2 fibrosis, with blast cells <5%. New DIPSS score calculated was Intermediate-2. Treatment with ruxolitinib 20 mg two times a day was started, with a partial response achieved after six months of therapy (improved Hb value over 10 g/dL, but under the normal limit, and resolution of the disease’s symptoms). With this response, a follow-up was initiated, not considering at the moment the option of a hematopoietic stem cell transplant (HSCT). This choice was supported by using the recently developed and validated prognostic model for the early response to ruxolitinib after 6 months (RR6 risk class Intermediate) [5,6,7]. After two more years of JAKi treatment, the patient was referred to our department with progressive anemia transfusion dependent, increased WBC value (about 29,000/mmc, with 25,000/mmc lymphocytes), reduction of platelets count (<20,000/mmc), and high LDH (3 times the upper limit of normal). Clinically, she reported dyspnea with a reduction of cardiac EF (30%), recurrent fever, several daily diarrheal discharges, and large cutaneous hematomas with diffuse skin ulcer-like lesions. In brief, clinical conditions had significantly worsened in the last 4 months. The patient was hospitalized, and ruxolitinib was stopped due to thrombocytopenia, with platelets below 15,000/mmc. For the suspicion of acute myeloid leukemia shift, a bone marrow study was performed, with the evidence of cellularity near 100%, severe dysplasia, however with CD34+ blast cells <5%. Grade 2 diffuse reticulin fibrosis was again confirmed, and unexpectedly reactive lymphocyte aggregates and numerous histiocytes containing amastigotes/Leishman-Donovan bodies were reported (Figure 1). The diagnosis of visceral leishmaniasis (highly suspicious at this point due to systemic clinical manifestations, with skin, cardiac and intestinal involvement) was confirmed through the detection of *Leishmania* kinetoplastic DNA (kDNA), in serum and bone marrow biopsy. A heart MRI was performed, showing a wide strip of late intramyocardial impregnation, associated with subtle enhancement of the pericardium and the effusion flap, possibly related to myopericarditis. Based on FDA guidelines, the patient was treated with liposomal amphotericin B 4 mg/kg daily on days 1–5, then weekly for 5 following doses [8].

At 21 days, the patients achieved a response to the aforementioned treatment, with the improvement of clinical conditions, an increase of heart EF to 50% and absence of dyspnea, the disappearance of diarrhea events. Interestingly, no copies of *Leishmania*-specific DNA were detected in peripheral blood together with a progressive increment of specific antibodies, despite previous ruxolitinib immunosuppressive therapy. The complete blood count showed an improvement in Hb (9 g/dL), and platelets (about 30,000/mmc), despite persistent lymphocytosis. The PCR study of peripheral blood samples showed the absence of *Leishmania*-specific DNA beyond +72 days from the start of liposomal amphotericin B treatment during the follow-up, guaranteeing the possibility for the patient to continue her therapeutic process with HSCT, at that moment the best choice for cytopenic myelofibrosis, without indication for ruxolitinib treatment due to thrombocytopenia.

## 3. Discussion

Ruxolitinib is the progenitor of a class of JAK inhibitor drugs (JAKi) that have changed the therapeutic approach and management of various diseases, in particular myelofibrosis. The JAK kinase pathways are required to activate the Src-kinase, the Ras-MAP kinase, the PI3K-AKT, and STAT signaling. The normal cellular activity could be disrupted by functional alterations in JAK pathways, leading to tumorigenesis in case of over-expression, as in the case of MF. On the other side, loss of JAK function (or the reduction given using JAK inhibitor drugs, such as ruxolitinib) lead to conditions characterized by severe-combined immunodeficiency [9]. The immunosuppressive potential, on the one hand, is widely used in the second-line therapy of steroid-refractory graft versus host disease (GVHD) in the setting of patients undergoing allogeneic bone marrow transplant (HSCT) [10]. On the other hand, it increases the risk of opportunistic infections. Indeed, infections are one of the leading causes of morbidity and mortality in MF patients, representing the reason for death in around 10% of the cases [11]. In fact, ruxolitinib interferes with the signaling of several cytokines and growth factors that play a pivotal role in the immune process, reducing pro-inflammatory cytokines in MF patients through a down-regulation of T-regs and impairment of the dendritic and natural killer (NK) cell function [12]. Immunosuppression due to this drug in MF patients was clearly seen in phase 3 studies (COMFORT-I and COMFORT-II). Mainly, bacterial infections that involve urinary tract infections, pneumonia, and sepsis, followed by viral agents, were reported. Focusing on the latter, the human herpesvirus-3 and influenza virus infections are the more frequent. These complications may be clinically relevant and able to delay the therapeutic process [13,14]. Then, opportunistic infections are to be considered. In the literature [15] the most frequent were tuberculosis, hepatitis B reactivation, cryptococcal, and pneumocystis jirovecii infections [12]. However, to our knowledge, very little data are present concerning parasite infections in myelofibrosis patients.

Several parasites are accountable for life-threatening conditions in profound immunocompromised subjects with impaired mediated-cell-mediated immunity. In AIDS patients, opportunistic infections are highly prevalent in those subjects with CD4 lymphocyte counts <200/mm^3^, with a prevalence of intracellular protozoa. In immunocompromised hosts, severe parasitic infections either arise from the reactivation of a previously contracted infection (toxoplasmosis or visceral leishmaniasis) or have a more severe course [16,17,18]. Leishmaniasis is an infectious disease of parasitic origin caused by protozoa of the genus *Leishmania*, an obligate intracellular parasite of the reticulo-histiocytic system of man and other mammals. Leishmaniasis is a vector-borne disease that is transmitted by the bite of infected female phlebotomine sandflies. In humans, it has three main forms: the cutaneous, the visceral (VL, or kala-azar), and the mucocutaneous. In particular, in immunosuppressed patients, the diagnosis is possible through polymerase chain reaction (PCR) on bone marrow and peripheral blood samples, with high sensitivity and specificity [19]. Moreover, it has been demonstrated that the sensitivity of PCR tests in peripheral blood samples can be superimposed on the direct examination of bone marrow aspirates, including in immunosuppressed patients [20]. In the VL form, characteristic symptoms may be irregular fever, hepatosplenomegaly, and pancytopenia associated with polyclonal hypergammaglobulinemia [21]. In addition to these classic symptoms, other manifestations involving different systems are variously reported. As in the case we described above, there may be cardiac involvement, manifesting as cardiomyopathy that can range from asymptomatic to rapidly impaired ventricular systolic function, with decompensated heart failure and cardiogenic shock. Cardiac tamponade is the major risk in case of pericarditis [22]. Although not yet understood in humans, the underlying pathophysiological mechanisms have been studied in the dog, where the development of an inflammatory reaction with diffuse areas of the myocardium being infiltrated with immune cells was seen [23]. Other manifestations are related to gastrointestinal involvement, with several diarrheal episodes that cannot otherwise be explained [24].

Once the human infection has occurred, the main defense mechanism against these protozoa able to survive inside the macrophages is constituted by cell-mediated immunity and, particularly, by the activation of the macrophages by cytokines produced by Th1 lymphocytes. In particular, resistance to protozoal infection is given by the production of IFNy and TNFα, promoting the elimination of *Leishmania* by macrophages through the production of nitric oxide (NO) and reactive oxygen species (ROS), efficacious in eradicating intracellular amastigotes [25,26]. However, naive CD4+ T-lymphocytes can also differentiate into Th2 lymphocytes, producing suppressive cytokines (IL-4, IL-10, and IL-13), leading to increased parasite survival and exacerbation of tissue damage [27]. In the case of immunosuppressed patients, the immune response may not be as effective. The T-cell lymphocytes of such patients are usually particularly impacted, due to distinct mechanisms; among these, of key importance is lymphocyte depletion, interference with cell maturation, cell cycling, and co-stimulation, simultaneously with induction of tolerance. All these circumstances can thereby modify the mechanisms of defense against microorganisms localized in the cells [16].

In our patient, considering the different cytokine patterns due to myelofibrosis and ruxolitinib treatment, we evaluated the cytokine oscillations expressed by CD4+ and CD8+ T-lymphocytes (IFNγ, TNFα, IL-2, IL-4, IL-17) at diagnosis of leishmaniasis and during treatment with amphotericin-B (Table 1).

At the time of diagnosis, IFNγ, TNFα and IL-2 levels on both CD4+ and CD8+ lymphocytes were low, contrarily to IL-4 expression, evidencing a lymphocyte redirection towards the immunosuppressive Th2 phenotype. Together with this, the critical decline in clinical conditions and the patient’s inability to eradicate or at least contrast the parasitic infection was observed. After the start of liposomal amphotericin-B, simultaneously with the decrease of *Leishmania*-specific DNA from the circulation, a lymphocytic shift in the Th1 class occurred, with a considerable increase in proinflammatory cytokines and a consequent immune capacity to eradicate the infection. This aspect was confirmed by PCR negativity after 14 days after the initial diagnosis. Curiously, after about six weeks since diagnosis of leishmaniasis and in a remission phase from the infectious disease, we measured a new reduction in TNFα and IFNγ. Thus, the inflammation picture with raised pro-inflammatory cytokines in the frame of MF, followed by a mismatched allocation of T-cell subsets during ruxolitinib treatment [28]. Furthermore, T-cell exhaustion defined by insufficient effector function, with a transcriptional state different from that of functional or memory T-cells and expression of inhibitory molecules [29], has to be taken into consideration. This process culminates in the consumption of the cells deputed to immune response [29]. While this effect can be exploited in preventing cytokine storms during T-cell involving therapies (such as chimeric antigen receptor T-cells and T-cell bispecific antibodies) [30] and during hyperinflammation syndromes [31], it can favor the progression of infectious diseases by preventing a normal immune response.

In the field of parasitic infections, the safety and tolerability of ruxolitinib combined with artemether-lumefantrine for the treatment of endemic forms of malaria, has been suggested. In fact, the inhibition of type I IFN exerted by ruxolitinib could prevent the dysfunctional immune response caused by *Plasmodium*, reducing the risk of recurrent infection or severe disease [32]. Recently, it has been reported a single patient developed haemophagocytic lymphohistiocytosis (HLH) secondary to *Leishmania* infection [33]. The treatment was based on a combination of amphotericin-B and ruxolitinib. The latter drug was introduced to counteract the cytokine storm caused by HLH. The association of the two drugs proved to be feasible and the patient fully recovered. In our case, the patient was already treated with ruxolitinib when amphotericin B was started; however, also in our hands, the association of the two drugs was easy to deliver. Furthermore, both clinical cases point to [34] the possible double role of ruxolitinib, combating the disease and the underlying immune system alterations on one side and reducing the immune response with an increased infectious risk on the other side. In this respect, the dose and timing of ruxolitinib could be critical.

As a final consideration, it is interesting to analyze the patient antibody production following infection. In fact, immunosuppression against the T-lymphocyte line is well known, while documenting the impact of JAK inhibition on antibody production is challenging to predict. Be sure an adequate B cell activity for antibodies production requires an elaborate interplay of intakes from different T-cells, including T-effector cells and T-regs [33]. Moreover, in MPNs the mutant clone could involve lymphoid-derived cells, thus causing possible abnormalities in B cells [35,36]. In addition, an abnormal B-cell response is documented in myeloproliferative neoplasm patients receiving ruxolitinib. Recent data in these patients show an impaired antibody response following SARS-CoV-2 dose vaccination [35,37].

Our patient was unable to eliminate the parasite, although a slight production of indirect fluorescent antibody test (IFAT) detectable IgG antibodies against *Leishmania* was detectable. On the contrary, when the therapy was suppressed, higher serum-specific IgG antibody titer against *Leishmania* antigens was detected. IFAT raised from 1:640 at diagnosis to 1:2560 about three months after the end of therapy with ruxolitinib (IFAT positive ≥ 1:80). Of course, the implementation of specific therapy with amphotericin-B liposomal had a positive effect on the elimination of the parasite DNA, as confirmed by negative PCR after only 14 days.

There are no robust data on antibody response in healthy subjects affected by VL, while there is little evidence in the cutaneous form. In an epidemiological study in endemic regions [36], antibody levels remained high in untreated patients while in treated patients the level of antibodies was inversely correlated with the duration of therapy. On the contrary, in our patient with VL we observed an increase in antibodies following the end of therapy, probably due to ruxolitinib withdrawal.

## 4. Conclusions

The immunosuppressive treatments in the setting of hematological diseases can confer an infectious risk able to complicate the patient outcome and sometimes be difficult to identify, with the chance of misunderstanding the diagnosis. This, in turn, could mislead the clinical suspect toward a not true MPN progression. In this setting, the underlying immunological aspects are to be adequately studied and understood to guarantee the best treatment for both the infectious and the primary disease.

## Figures and Tables

**Figure 1 jcm-12-00578-f001:**
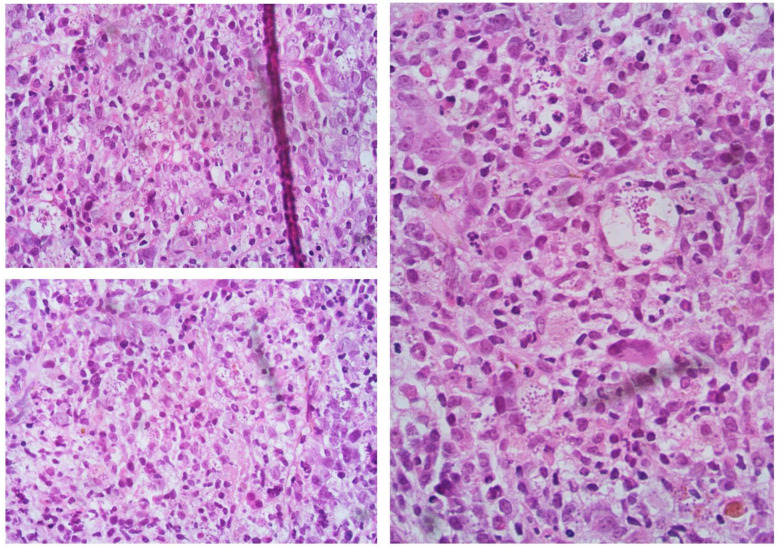
Evidence of amastigotes/Leishman-Donovan bodies in bone marrow biopsy, as described in the text.

**Table 1 jcm-12-00578-t001:** Evidence of fluctuation in cytokines values at three different time-point during the follow-up after treatment.

	T-CD4+				T-CD8+			
	INFγ+	TNFα+	IL-2+	IL-4+	IL-17+	INFγ+	TNFα+	IL-2+
At diagnosis	7.01	32.5	21.2	6.36	3	16.75	17.6	9.4
+21	25.2	75	63.4	3.28	2.11	56	63.4	41
+42	18.7	58.7	52.8	0.6	1.64	42.5	48	34.3

## Data Availability

Data are stored in a data worksheet and can be made available upon a request.

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
