# Peer review of "Leishmania Infection during Ruxolitinib Treatment: The Cytokines-Based Immune Response in the Setting of Immunocompromised Patients"

_jcm, 2023, doi:10.3390/jcm12020578_

Round 1
Reviewer 1 Report
This is indeed the first reported case with leishmaniasis in a patient with MF after many years of treatment with ruxolitinib.
ad point 2 - case presentation: The disease should be described in more detail, I think additionally to IPSS DIPSS and DIPSS plus should be reported for the timepoint when ruxo was started. Was there a bone marrow before starting ruxo? Why was the patient not moved forward to allo transplant which was mentioned in line 92 as s life-saving procedure for the future.
ad point 3 - discussion: overall informative, well written
more information about published cases with opportunistic infections, reactivations of infections, which are available in the literature, should be added.
Author Response
Herein, we are pleased to resubmit the manuscript entitled “Leishmania infection during ruxolitinib treatment: the cytokines-based immune response in the setting of immunocompromised patients”.
Thank you for your suggestions during the review phase.
Specifically, we enclose a table below in which we respond point by point to the notes raised during the review phase.
Moreover, we practised a deep review of the English language with the supervision of an English University service.
Thank you in advance,
Sincerely,
Dr. Andrea Duminuco
Reviewer observation |
Author response |
The disease should be described in more detail, I think additionally to IPSS DIPSS and DIPSS plus should be reported for the timepoint when ruxo was started. Was there a bone marrow before starting ruxo? Why was the patient not moved forward to allo transplant which was mentioned in line 92 as s life-saving procedure for the future. |
Thank you for your suggestion. We added the bone marrow evaluation before the start of ruxolitinib treatment, with a DIPSS prognostic score. Unfortunately, DIPSS plus cannot be used due to the unavailability of the karyotype, as each bone marrow examination has failed due to a dry tap. Concerning HSCT, we removed the term “life-saving”. The patient was referred to transplant procedure because the patient could no longer undergo therapy with ruxolitinib as she was thrombocytopenic (<50,000/mmc). We further reinforced our choice not to initiate the patient immediately for transplantation also based on the early response score to ruxolitinib RR6, according to which the patient was at intermediate risk, with an mOS of 61 months |
More information about published cases with opportunistic infections, reactivations of infections, which are available in the literature, should be added |
Thank you for your suggestion. We inserted a discussion and literature review of opportunistic infections in this patient setting |
Reviewer 2 Report
The case is interesting and well worth publishing. Some of the sentences are very long. It would be an improvement of the content if some of them were shortened. The conclusion should be reformulated and improved.
It could be an advantage to include the reference "Infect Drug Resist by Cui T. et al., 2022 (Nov.)" PMID: 36386416: The treatment Based on Ruxolitinib and Ampho B having an effect on relapsed Leichmaniasis-related HLH, and explain this in relationship to the case in this manuscript.
Author Response
Herein, we are pleased to resubmit the manuscript entitled “Leishmania infection during ruxolitinib treatment: the cytokines-based immune response in the setting of immunocompromised patients”.
Thank you for your suggestions during the review phase.
Specifically, we enclose a table below in which we respond point by point to the notes raised during the review phase.
Moreover, we practised a deep review of the English language with the supervision of an English University service.
Thank you in advance,
Sincerely,
Dr. Andrea Duminuco
Reviewer observation |
Author response |
Some of the sentences are very long. It would be an improvement of the content if some of them were shortened. |
Thank you for your indication. We practised a deep review of the English language with the supervision of an English University service.
|
The conclusion should be reformulated and improved. |
Thanks for your suggestion. We reformulated the conclusion, improving the final message |
It could be an advantage to include the reference "Infect Drug Resist by Cui T. et al., 2022 (Nov.)" PMID: 36386416: The treatment Based on Ruxolitinib and Ampho B having an effect on relapsed Leichmaniasis-related HLH, and explain this in relationship to the case in this manuscript.
|
It is a interesting report. We inserted this paper in the discussion, explaining the role of ruxolitinib as risk factor (such as in our case) or potential curative role (in the case of HLH reported) |